# Protective Effect of Ferulic Acid against Hydrogen Peroxide Induced Apoptosis in PC12 Cells

**DOI:** 10.3390/molecules26010090

**Published:** 2020-12-28

**Authors:** Hironao Nakayama, Masako Nakahara, Erina Matsugi, Midori Soda, Tomoka Hattori, Koki Hara, Ayuki Usami, Chiaki Kusumoto, Shigeki Higashiyama, Kiyoyuki Kitaichi

**Affiliations:** 1Department of Medical Science and Technology, Hiroshima International University, Higashi-hiroshima, Hiroshima 739-2695, Japan; nakahara@hirokoku-u.ac.jp (M.N.); sm20210@ms.hirokoku-u.ac.jp (E.M.); kusumoto@hirokoku-u.ac.jp (C.K.); 2Laboratory of Pharmaceutics, Gifu Pharmaceutical University, Gifu 501-1196, Japan; soda@gifu-pu.ac.jp (M.S.); 155058@gifu-pu.ac.jp (T.H.); 165063@gifu-pu.ac.jp (K.H.); 175022@gifu-pu.ac.jp (A.U.); 3Division of Cell Growth and Tumor Regulation, Proteo-Science Center, Ehime University, Toon, Shitsukawa, Ehime 791-0295, Japan; shigeki@m.ehime-u.ac.jp

**Keywords:** oxidative stress, ferulic acid, apoptosis, cell signaling

## Abstract

Ferulic Acid (FA) is a highly abundant phenolic phytochemical which is present in plant tissues. FA has biological effects on physiological and pathological processes due to its anti-apoptotic and anti-oxidative properties, however, the detailed mechanism(s) of function is poorly understood. We have identified FA as a molecule that inhibits apoptosis induced by hydrogen peroxide (H_2_O_2_) or actinomycin D (ActD) in rat pheochromocytoma, PC12 cell. We also found that FA reduces H_2_O_2_-induced reactive oxygen species (ROS) production in PC12 cell, thereby acting as an anti-oxidant. Then, we analyzed FA-mediated signaling responses in rat pheochromocytoma, PC12 cells using antibody arrays for phosphokinase and apoptosis related proteins. This FA signaling pathway in PC12 cells includes inactivation of pro-apoptotic proteins, SMAC/Diablo and Bad. In addition, FA attenuates the cell injury by H_2_O_2_ through the inhibition of phosphorylation of the extracellular signal-regulated kinase (ERK). Importantly, we find that FA restores expression levels of brain-derived neurotrophic factor (BDNF), a key neuroprotective effector, in H_2_O_2_-treated PC12 cells. As a possible mechanism, FA increases BDNF by regulating microRNA-10b expression following H_2_O_2_ stimulation. Taken together, FA has broad biological effects as a neuroprotective modulator to regulate the expression of phosphokinases, apoptosis-related proteins and microRNAs against oxidative stress in PC12 cells.

## 1. Introduction

Oxidative stress-mediated cellular injury has been implicated in various diseases, such as cancer [1], cardiovascular diseases [2], and neurodegenerative diseases including Alzheimer’s and Parkinson’s diseases [3]. A basal level of reactive oxygen species (ROS) is indispensable for the manifestation of cellular functions, whereas increased formation of ROS causes damage to cellular macromolecules such as DNA, lipids and proteins, eventually leading to necrosis and apoptotic cell death. It has been shown that ROS are involved in nervous system dysfunction and brain disorders. After brain injury, cellular functions are impaired by the excess production of free radicals, which are generated through several different cellular pathways [4,5]. Therefore, the scavenging of ROS mediated by antioxidants may be a potential strategy for retarding the diseases’ progression. Many synthetic antioxidants have potential as strong radical scavengers; however, they are also carcinogenic and cause liver damage [6]. On the other hand, the exogenous consumption of antioxidants from natural sources, including plant, animal, and mineral, can produce beneficial effects on human health and reduce the incidence of free radical-induced diseases, including neurodegenerative disorders. For this reason, much attention has been focused on the therapeutic use of antioxidants from natural sources with neuroprotective potential [7,8].

Ferulic acid (4-hydroxy-3-methoxycinnamic acid, FA) is a widely distributed phenolic compound in plant tissues (e.g., seed plants, vegetables and fruits), therefore, FA is one of the most abundant phenolic compounds in the human diet [9,10]. FA exhibits several physiological functions, for example, it has both anti-inflammatory and antioxidant properties due to its phenolic nucleus and an extended side chain [9,10]. FA readily forms a resonance stabilized phenoxy radical which accounts for its potent antioxidant potential [11]. It has been demonstrated that FA exerts protective effects against the impaired learning and memory induced by ischemia reperfusion in vivo by antioxidant and anti-apoptotic mechanisms [5]. Moreover, oral FA treatment into Alzheimer’s disease model mouse reduces amyloidogenic amyloid β-protein precursor metabolism by modulating β-secretase [12,13], suggesting that the free radical scavenger activity of FA is a promising compound against neurodegenerative disease, such as Alzheimer’s disease [9,14]. In addition to the above bioactivities, the antioxidant effect of FA has been verified against several acute and chronic pathologies, including cancer [15], cardiovascular disease [16], and diabetes [17]. Several studies have reported that encapsulation of FA into a proper drug delivery system enhances FA bioactivities because of low solubility, low stability, and short residence time [18,19]. Carbone and co-workers demonstrated that the combined delivery of FA and *Lavandula* essential oils promotes cell proliferation and migration in would healing [20]. Although it is clear that FA has therapeutic potential to treat or prevent a wide variety of diseases, the detailed molecular mechanisms involved are not well understood.

Hydrogen peroxide (H_2_O_2_) is thought to be the major precursor of ROS and is utilized extensively as an inducer of oxidative damage to interpret molecular mechanisms and therapeutic potential of antioxidants. Rat pheochromocytoma PC12 cells are a well-known model for studying neuronal signaling pathway and neuronal functions [21]. For example, H_2_O_2_ induces cytotoxicity in PC12 cell and alters apoptosis-related proteins, including anti-apoptosis proteins, pro-apoptosis proteins, and caspases [22,23]. In this study, we show that the treatment of PC12 cells with FA prior to H_2_O_2_ exposure effectively inhibits cell apoptosis. Furthermore, FA decreases the intracellular ROS, pro-apoptotic proteins, SMAC/Diablo and Bad, and inhibits the MAPK signaling pathway. These results demonstrate that FA is promising as a potential therapeutic candidate for neurodegenerative diseases resulting from oxidative damage and further research on this topic should be encouraged.

## 2. Materials and Methods

### 2.1. Antibodies and Reagents

The following antibodies were used in this study: the rabbit monoclonal anti-smac/diablo antibody (#15108), rabbit monoclonal anti-bad (#9068), rabbit polyclonal anti-phospho-Erk1/2 (Thr202/Tyr204) antibody (#9101) and mouse monoclonal anti-Erk1/2 antibody (#4696) were purchased from Cell Signaling Technology (Danvers, MA, USA). Mouse monoclonal anti-β-actin antibody (AC-15) were purchased from Sigma-Aldrich (St. Louis, MO, USA). The rabbit monoclonal anti-BDNF antibody (ab108319) was purchased from abcam. Trans-ferulic acid (FA) was obtained from Sigma-Aldrich. FA was dissolved in culture medium at 10 mM and then used for in vitro assay. Actinomycin (ActD) was purchased from Wako (Osaka, Japan). ActD was dissolved in ethanol at 5 mg/mL and then diluted in culture medium.

### 2.2. Cell Culture

Rat pheochromocytoma PC12 cells were purchased from ATCC (CRL-1721). Cells are grown in RPMI 1640 supplemented with 5% heat-inactivated fetal bovine serum (FBS) and 10% horse serum (HS). Cells were cultured in a humidified incubator at 37 °C/5% CO_2_.

### 2.3. Apoptosis Assay

Apoptosis was detected using a commercially available kit (Annexin V-FITC kit; MBL, Nagoya, Japan) in accordance with the protocol provided by the manufacturer. PC12 cells were treated with FA (40 µM) in the serum-reduced (2% FBS and 10% HS) RPMI1640 medium overnight. Then, cells were treated with H_2_O_2_ (1 mM) or actinomycin (ActD, 20 μM) for 1 h. H_2_O_2_-treated cells were stained with Annexin V-FITC and PI for 15 min at room temperature and subsequently analyzed via flow cytometry (BD Accuri C6 plus, BD Biosciences, San Jose, CA, USA). Apoptotic cell population was analyzed by the upper right quadrant. ActD-treated cells were stained with 7AAD (Bio-Rad, Hercules, CA, USA) for 10 min and subsequently analyzed via flow cytometry.

### 2.4. ROS Assay

ROS was detected using a commercially available kit (Cell Meter Fluorimetric Intracellular Total ROS Activity Assay Kit, AAT Bioquest, Sunnyvale, CA, USA) in accordance with the protocol provided by the manufacturer. Cells were incubated with Amplite ROS Green for 1 h and H_2_O_2_ (1 mM) for 30 min. The cells were measured using the flow cytometry using the filters for Ex/Em = 490/520 nm.

### 2.5. Proteome Profiler Antibody Array

The Human Phospho-Kinase Array Kit (ARY003) and Proteome Profiler Human Apoptosis Array Kit (ARY009) were obtained from R&D Systems. PC12 cells were treated with FA (40 μM) overnight. Cell lysates were collected at 30 min after H_2_O_2_ (0.5 mM) treatment and the levels of phospho-or apoptosis-related proteins were analyzed with these arrays, according to the manufacturer’s instructions. The intensity of each dot was measured using Image J software. The ratios indicated were calculated by using the intensities of the corresponding protein dots after background correction and normalization of the intensities according to the mean of the positive controls (Appendix A). N/D indicates “not detected” in the paper.

### 2.6. Western Blotting

Samples were separated by SDS-PAGE and transferred to a nitrocellulose membrane. The membranes were blocked with 4% skim milk in TBS-T (0.05% Tween-20 in TBS) for 30 min, which was followed by incubation with primary antibodies. After being washed with TBS-T, the membranes were incubated with the appropriate horseradish peroxidase (HRP)-conjugated secondary antibodies. Immunoreactivity was detected by using enhanced chemiluminescence detection reagents.

### 2.7. Isolation of RNA and Quantitative RT-PCR

RNA was isolated using the RNeasy micro kit (Qiagen, Hilden, Germany), according to the manufacturer’s instructions. Reverse transcription of RNA was performed with TaqMan™ MicroRNA Reverse Transcription Kit, according to the manufacturer’s protocol. After first-strand synthesis, qRT-PCR was performed using the TaqMan™ Universal Master Mix II with a 7300 real-time PCR system (Applied Biosystems, Foster City, CA, USA). The microRNA-10b expression was analyzed using TaqMan™ Assays. TaqMan™ microRNA Control Assay (U87) was used for an internal control.

### 2.8. Statistical Analysis

All assays were independently performed three times. The results are represented as mean ± SEM. Analysis of variance (ANOVA) with Bonferroni post-hoc test was used for multiple comparisons. *p* < 0.05 was considered statistically significant.

## 3. Results and Discussion

To determine the effects of FA against oxidative stress, we initially pretreated PC12 cells with FA (40 µM) overnight and subsequently cultured cells in the absence or presence of H_2_O_2_ (1 mM) for another 45 min. The H_2_O_2_-induced cell apoptosis was assessed by Annexin V-FITC assays [24]. Treatment with H_2_O_2_ resulted in a marked induction of apoptosis of PC12 cells compared to control (Figure 1a). In contrast, pretreatment of the cells with FA had reduced levels of apoptosis induced by H_2_O_2_ (*p* < 0.05, Figure 1a). In addition, treatment with FA alone without H_2_O_2_ did not significantly affect apoptosis of PC12 cells compared to control. Of note, this effect of FA on the inhibition of apoptosis was similar to that observed when cells are treated with actinomycin D (ActD), a potent inducer of apoptosis (*p* < 0.001, Figure 1b). Importantly, when the cells are pretreated with H_2_O_2_ for 15 min and then treated with FA for another 30 min, FA failed to inhibit apoptosis of PC12 cells (Appendix A). These data rule out the possibility that FA reacted directly with H_2_O_2_. Altogether, these results suggest that FA is a potent inhibitor of apoptosis induced by oxidative stress.

Oxidative stress-induced ROS production causes cellular damage and apoptosis due to oxidation of many essential proteins [25,26,27]. Moreover, ROS has been known as a pro-apoptotic factor that activates stress-activated protein kinases. We evaluated the accumulation of ROS after H_2_O_2_ stimulation in PC12 cells using the fluorescence probe. The results showed that the level of ROS accumulation was higher in H_2_O_2_-treated cells. On the other hand, pretreatment with FA attenuated the fluorescence intensity in H_2_O_2_-treated PC12 cells (Figure 1c), suggesting that FA exerts its antioxidant effect in the intracellular compartment. A recent study showed that FA significantly decreased ischemia-induced apoptosis as well as ROS accumulation in PC12 cells, and increased the generation of the cellular antioxidants, superoxide dismutase, and glutathione peroxidase [5]. Therefore, these results confirm that FA has antioxidant activity against ROS.

To determine the effect of FA on H_2_O_2_-induced PC12 cell apoptosis, we profiled levels of apoptosis-related proteins in PC12 cells. We found that FA inhibited H_2_O_2_-induced apoptotic protein, notably, Bad (Figure 2a and Appendix A), which was confirmed by Western blot analysis (Figure 2b). ROS activates p53 and/or c-Jun N-terminal kinase (JNK), which activate pro-apoptotic Bcl-2 proteins, including Bad [27]. Bad is a member of the BH3-only family which is involved in initiating apoptosis. Under conditions of stress, Bad forms heterodimers with anti-apoptotic proteins such as Bcl-2, Bcl-XL, thus inhibiting their anti-apoptotic properties [27]. We also found that FA inhibited pro-apoptotic protein Bax as well (Appendix A). Thus, these data suggest that FA inhibits H_2_O_2_-induced PC12 cell apoptosis by regulating expression level of pro-apoptotic proteins (Figure 2a and Appendix A). In addition, FA suppressed SMAC/Diablo expression induced by H_2_O_2_ (Figure 2a and Appendix A). Excessive ROS production would damage mitochondrial membrane integrity and affect the energy production in mitochondria, resulting in mitochondrial dysfunction [27,28]. Furthermore, mitochondrial dysfunction includes a decrease in mitochondria membrane potential, activation of caspase-3, and apoptosis. SMAC/Diablo is a mitochondria-derived pro-apoptotic protein. In the response to diverse pro-apoptotic stimulation, SMAC/Diablo is released from mitochondria and enters the cytosol, possibly by neutralizing the caspase-inhibitory properties of the inhibitor of apoptosis proteins (IPA) family of proteins [29,30]. This leads ultimately to apoptotic cell death by both caspase-dependent and -independent mechanisms. Therefore, FA might change the occurrence of mitochondrial dysfunction after oxidative stress.

In order to further examine the effect of FA on intracellular signaling responses, we profiled levels of phosphokinases in PC12 cells. As a result, the phosphorylation of ERK by H_2_O_2_ was inhibited by pretreatment with FA (Figure 3a and Appendix A). By Western blot analysis, it was found that H_2_O_2_ stimulation activated ERK in a dose-dependent manner, whereas pretreatment with FA inhibited it (Figure 3b). Activation of ERK inhibits apoptosis in response to several stimulations, including tumor necrosis factor, Fas ligand, and H_2_O_2_ [31,32]. In contrast, ERK can function in a pro-apoptotic manner. For example, ERK activation contributes to 6-hydroxydopamine (6-OHDA)-induced dopaminergic neuronal cell death [33]. Moreover, persistent activation of ERK is associated with glutamate-induced oxidative toxicity in neuronal cells, and inhibition of ERK activation protects cells from glutamate toxicity [34]. Therefore, ERK contributes to pro-apoptotic signaling in neuronal cells. Together, these results demonstrate that FA inhibits H_2_O_2_-induced apoptosis through the regulation of intracellular ROS level, mitochondrial-dependent pathway, and MAPK pathway in PC12 cells.

It has been reported that FA upregulates the levels of BDNF, a neuroprotective effector, in the prefrontal cortex and hippocampus in chronic unpredictable mild stress model mice [35]. Therefore, we examined the effect of FA on expression of BDNF in H_2_O_2_-treated PC12 cells. Western blot analysis revealed that FA prevented the reduced protein expression of BDNF by H_2_O_2_ in PC12 cells (Figure 4a). Previous reports have shown that BDNF expression is directly regulated by several microRNAs, such as microRNA-1 (miR-1), miR-9, miR-10b, miR-155 and miR-191, since these microRNAs target 3′UTR of BDNF [36,37,38]. We found that the expression of miR-10b was induced by H_2_O_2_ in PC12 cells, whereas FA inhibited H_2_O_2_-induced miR-10b (Figure 4b). As a possible mechanism, FA increased the expression of BDNF through the inhibition of miR-10b induction following H_2_O_2_ stimulation, thereby protecting PC12 cells from apoptosis.

## 4. Conclusions

In summary, our data first demonstrate that FA exerts neuroprotective effect against oxidative cytotoxicity in PC12 cells by decreasing level of intracellular ROS and mitochondrial pro-apoptotic proteins, and regulating the MAPK pathway. Moreover, FA increases the expression of BDNF in PC12 cells by possibly regulating microRNAs expression induced by oxidative stress. Further studies are needed to identify FA-targeting microRNAs to completely understand the mechanism of the neuroprotective effect of FA.

## Figures and Tables

**Figure 1 molecules-26-00090-f001:**
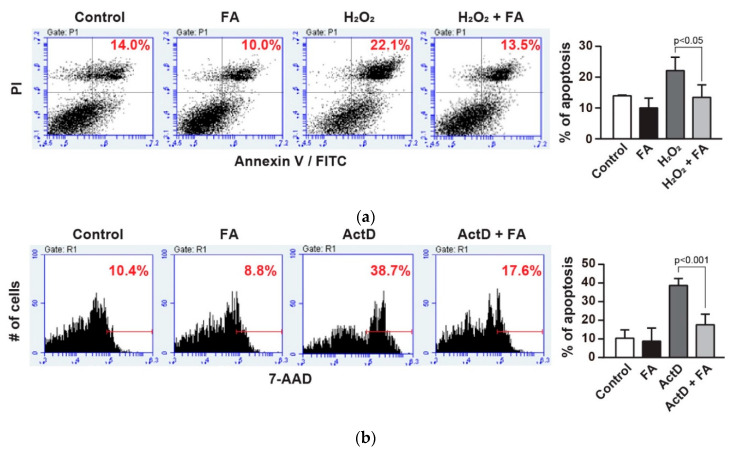
Effect of ferulic acid (FA) against oxidative stress in PC12 cells. (**a**) PC12 cells were pretreated with FA (40 μM) overnight and subsequently cultured cells in the absence or presence of H_2_O_2_ (1 mM) for another 30 min. Cells were stained with Annexin V-FITC/PI to detect apoptotic cells (upper right quadrant) by flow cytometry. (**b**) PC12 cells were pretreated with FA (40 μM) overnight and subsequently cultured cells in the absence or presence of ActD (20 µM) for another 1 h. ActD-treated cells were stained with 7AAD for 10 min and subsequently analyzed by flow cytometry. (**c**) PC12 cells were pretreated with FA (40 μM) overnight and subsequently incubated with Amplite™ ROS Green for 1 h. Then, cells were cultured in the absence or presence of H_2_O_2_ (1 mM) for another 30 min, and were analyzed by flow cytometry. Data represent the mean ± SD.

**Figure 2 molecules-26-00090-f002:**
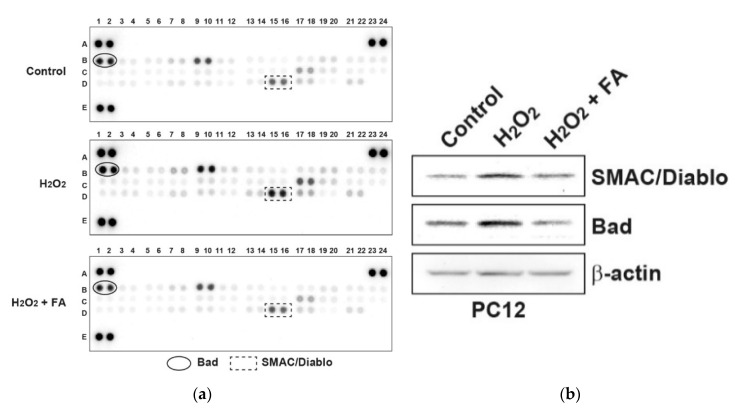
FA inhibits H_2_O_2_-induced apoptosis-related proteins in PC12 cells. (**a**) PC12 cells were pretreated with FA (40 μM) overnight and subsequently cultured cells in the absence or presence of H_2_O_2_ (0.5 mM) for another 30 min. Cell lysates were evaluated by apoptosis-related proteins antibody array according to the manufacturer’s instructions. The intensity of each dot/phosphoprotein was measured using Image J software. (**b**) Results of the array were validated by Western blot analysis.

**Figure 3 molecules-26-00090-f003:**
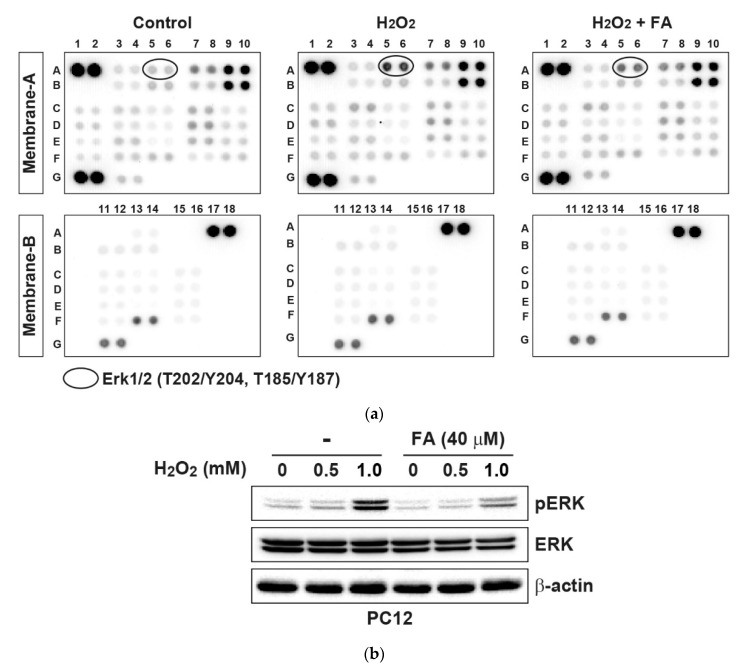
FA inhibits H_2_O_2_-induced phosphokinases in PC12 cells. (**a**) PC12 cells were pretreated with FA (40 μM) overnight and subsequently cultured cells in the absence or presence of H_2_O_2_ (0.5 mM) for another 30 min. Cell lysates were evaluated by phosphoprotein kinase antibody array according to the manufacturer’s instructions. The intensity of each dot/phosphoprotein was measured using Image J software. (**b**) Results of the array were validated by Western blot analysis.

**Figure 4 molecules-26-00090-f004:**
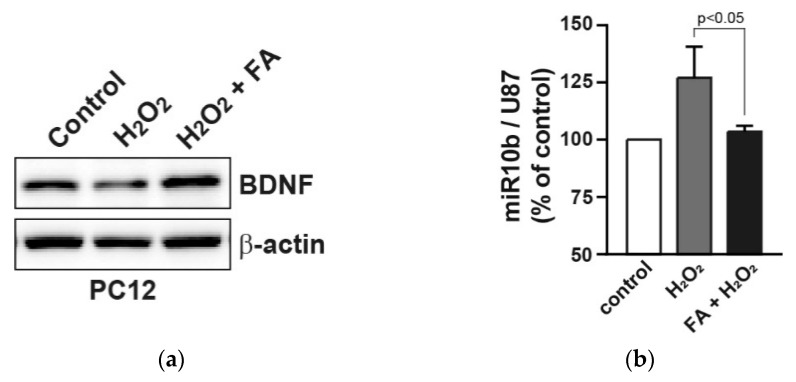
FA increases the reduced BDNF expression by H_2_O_2_ in PC12 cells. (**a**) PC12 cells were pretreated with FA (40 μM) overnight and subsequently cultured cells in the absence or presence of H_2_O_2_ (25 μM) for 24 h. Cell lysates were analyzed by Western blot analysis. (**b**) Expression levels of miR-10b were measured by qRT-PCR. miR-10b levels were normalized to U87 microRNA. Data represent the mean ± SD.

## Data Availability

Data is contained within the article or supplementary material.

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
