# Peer review of "Protective Effect of Ferulic Acid against Hydrogen Peroxide Induced Apoptosis in PC12 Cells"

_molecules, 2020, doi:10.3390/molecules26010090_

Round 1

Reviewer 1 Report

The paper “Protective Effect of Ferulic Acid against Hydrogen Peroxide-induced Apoptosis in PC12 cells” reports the investigation of the mechanisms of action of ferulic acid (FA). The Authors found that FA inhibited H2O2-induced apoptosis by attenuating the mitochondria-mediated response by inhibiting the MAPK pathway. The paper is well written and organized, and the results are interesting.

Some points should be clarified before publication:

- In the introduction, the shortcomings of ferulic acid (i.e., poor solubility and stability, short residence time) should be discussed, along with possible strategies to solve this problem through its formulation in innovative delivery systems: see for example “Ferulic Acid-NLC with Lavandula Essential Oil: A Possible Strategy for Wound-Healing? Nanomaterials 2020, 10, 898”.

- FA was used at a 40 microM concentration: in what solvent was it dissolved?

- Lines 149-150: the Authors report “FA also attenuated H2O2-induced apoptosis in human umbilical vein endothelial cell (data not shown).” If the results are not shown, there is no need to report this sentence. Besides, this is the only reference to tests on umbilical cells in the whole manuscript.

- The novelty of the work should be highlighted.

Reviewer 2 Report

Although this manuscript provides some interesting scientific results several deficiencies should be addressed before acceptance for publication in the Molecules.

There are several research reports related to this study. Please clarify the novelty of this study.

Ferulic acid inhibits H2O2-induced oxidative stress and inflammation in rat vascular smooth muscle cells via inhibition of the NADPH oxidase and NF-κB pathway

Cao, Yan-jun; Zhang, Yan-min; Qi, Jun-peng; Liu, Rui; Zhang, Han; He, Lang-chong

International Immunopharmacology (2015), 28(2), 1018-1025.

Please describe the source of the PC12 cell line in the paper.

Can these results rule out that ferulic acid reacted directly with hydrogen peroxide and actinomycin D? Did you conduct an experiment in which ferulic acid was added after treatment with hydrogen peroxide or actinomycin D?

Reviewer 3 Report

The manuscript "Protective Effect of Ferulic Acid against Hydrogen 2 Peroxide-induced Apoptosis in PC12 cells" is well written and shows an interesting result to regulation or stimulation of the expression of protein during oxidative stress induced by H2O2. 

The authors analyzed the effect of ferulic acid (FA) during apoptosis caused by H2O2 and observed the attenuating of the mitochondria-mediated response by inhibiting the MAPK pathway

The conclusion is that FA could be a neuroprotective effect against oxidative stress-related diseases, such as Alzheimer’s disease.

FA modulates apoptotic proteins like BAD and Bcls (Table 1) and phosphokinase protein showed in table 2.

Minor review:

 I felt a lack of detailed explanation of the results in the figure legend. Please consider reviewing the legend of Figure 2 and Figure 3 including the description of the main results are welcome to direct view for the readers.

Round 2

Reviewer 2 Report

There are no additional comments.